# The Removal of Inclusions with Different Diameters in Tundish by Channel Induction Heating: A Numerical Simulation Study

**DOI:** 10.3390/ma16155254

**Published:** 2023-07-26

**Authors:** Bing Yi, Guifang Zhang, Qi Jiang, Peipei Zhang, Zhenhua Feng, Nan Tian

**Affiliations:** 1Faculty of Metallurgical and Energy Engineering, Kunming University of Science and Technology, Kunming 650093, China; yibing8578@163.com (B.Y.); zpei861@163.com (P.Z.); fengzhenhua666@126.com (Z.F.); tian1852558@163.com (N.T.); 2Huanan Zhongke Electric Co., Ltd., Electromagnet Center, Yueyang 414000, China

**Keywords:** tundish metallurgy, dual-channel induction heating, inclusion removal, numerical simulation

## Abstract

The quality of the bloom will be impacted by the non-metallic impurities in the molten steel in the tundish, which will reduce the plasticity and fatigue life of the steel. In this research, a mathematical model of a six-flow double-channel T-shaped induction heating tundish was established, the effects of induction heating conditions on the removal of inclusions in the tundish were investigated, and the impact of various inclusion particle sizes on the removal effect of inclusions under induction heating was explored. The results show that the Residence Time Distribution (RTD) curve produced through numerical simulation and physical simulation is in good agreement. The reduction of inclusion particles in the channel is made affordable by the dual-channel induction heating technique. As the diameter of inclusion particles increases from 10 μm to 50 μm, the probability of inclusion particles being removed from the channel gradually decreases from 70.9% to 56.1%.

## 1. Introduction

The quality of the bloom is significantly impacted by the non-metallic impurities in the molten steel in the tundish, and the control of inclusions has always been an important direction of iron and steel metallurgy [1,2]. The number, size, chemical content, shape, and homogeneity of inclusions in the tundish can all be controlled. The behaviors of inclusions mainly include floating, collision condensation, and adsorption [3,4]. Moreover, after turbulence controllers, baffles and dams are added in the tundish, the extended residence time of the molten steel in the tundish is conducive to the full flow of the molten steel in the tundish, so the removal effect of the inclusions in the tundish will be significantly improved [5,6]. However, the temperature of molten steel will gradually decrease with the flow of molten steel, and the viscosity will increase, which is not conducive to the flow and removal of inclusions in the tundish. Wang et al. [7] and Xing et al. [8] reported that the electromagnetic induction heating technology could effectively solve the above problem and can realize the pouring of molten steel at a low degree of superheat, which helps to avoid center segregation and center keyhole and improves the quality of the continuous casting bloom. However, adding a large number of flow control devices, such as dams, will increase the contact area between the molten steel and the refractory material, thereby polluting the molten steel and ultimately affecting the quality of the billet [9].

Based on the principle of electromagnetic mutual induction followed in the metallurgical process, the researchers have developed a dual-channel induction heating tundish, which can compensate for the temperature loss during the flow of molten steel and facilitate the removal of inclusions in molten steel [10]. Mabuchi et al. [11] have found that molten steel will flow obliquely upwards under the influence of buoyancy and electromagnetic force after passing through the induction heating channel, which will promote the full flow of molten steel in the tundish and facilitate the removal of inclusions. Wang et al. [12] studied the effect of induction heating power on the movement of inclusion particles by combining experiments and simulations, and found that the electromagnetic force acting on molten steel is helpful for the removal of inclusion particles, and the larger the particle size of the inclusions, the more obvious the removal effect. Although the thermophoretic force is not conducive to the removal of inclusion particles, when the induction heating power is 1200 kW, the thermophoretic force suffered by inclusion particles is less than 3%, which can be ignored. Lei et al. [13] studied the physical field of inclusions through the population conservation model or inclusion mass conservation model. The study found that under the action of an electromagnetic field, both the electromagnetic force and induction heating are helpful to the removal of inclusion particles, and the former plays a dominant role. The removal rate of inclusion particles in the tundish increased from 21.4% to 31.05% after using channel induction heating, and the removal rate of inclusion particles in the channel was 1/3 of the removal rate of inclusion particles in the entire tundish. Miki et al. [14] conducted a simulation study on the trajectory of non-metallic inclusions and found that the inclusions’ particle size significantly impacted the removal rate. Moreover, when the inclusions collide and grow up from small-grained to large-sized inclusions, it is possible for the small-grained inclusions to float up and then be separated. Additionally, the size of the inclusions in the induction heating tundish has an important influence on its removal effect, which has become a hot spot for metallurgy experts [15,16].

According to the research of the above-mentioned scholars and further literature research, the current research mainly includes the distribution of flow field, temperature field, and electromagnetic force, and the movement of inclusion particles in the two-flow induction heating tundish of the curved channel [17,18,19]; the influence of the application of double-channel and four-channel bag types on the temperature field of molten steel in the six-flow induction heating tundish, and the influence of different power on the removal of inclusion particles in the channel [20,21], etc. However, few scholars have reported on the influence of inclusions with different particle sizes in different regions of the tundish in the six-fluid induction heating tundish. At the same time, the distribution of inclusions in different regions has a certain influence on optimizing the flow field, improving the crystallinity of molten steel, and improving the quality of the bloom. Therefore, this article investigates the removal effect and trajectory of inclusions in the channel zone and the discharging chamber in the tundish with and without induction heating so as to provide a theoretical reference for tundish induction heating to remove inclusions.

## 2. Model Building

### 2.1. Physical Model

#### 2.1.1. Geometry

Figure 1 is a schematic diagram of the three-dimensional geometry of the dual-channel induction heating tundish and the arrangement of the induction coils. The tundish is suitable for pouring 380 × 280 mm^2^ billets, the casting speed is 0.68 m/min, and the working fluid level is 900 mm. The distance between the four sides of the iron core and the channel is equal. Relevant dimensions include: the length of the top of the receiving chamber is 3086 mm, the length of the bottom is 2863 mm, the length of the channel is 1630 mm, the diameter of the main channel and the diameter of the sub-channel are both 160 mm, the top length and bottom length of the discharging chamber are 8141 mm and 7824 mm, and the distance between adjacent water outlets is 1500 mm. Moreover, the diameters of the water inlet and outlet are 90 mm and 40 mm, respectively.

The molten steel enters the receiving chamber through the long nozzle and impacts the turbulence controller. Then, the molten steel flows fully into the receiving chamber to form a good flow field. At the same time, most of the inclusions in the molten steel are removed in the receiving chamber. The main removal method is that the inclusion particles float to the surface of the molten steel and are adsorbed by steel slag and adhered to the wall of the receiving chamber. The molten steel enters the channel area after fully flowing in the receiving chamber. At this time, due to induction heating and the Lorentz force, swirling flow is formed within the channel, and the temperature rises rapidly. Due to the effect of the Lorentz force, some inclusion particles will be adsorbed on the inner wall of the channel. The high-temperature molten steel flowing out of the channel forms a density difference with the molten steel in the discharging chamber, so the molten steel flows obliquely upward after flowing out of the channel, which helps the molten steel flow fully in the discharging chamber and promotes the removal of inclusion particles. Finally, the molten steel flows out through the outlet.

#### 2.1.2. Experimental Method

It is the most common method to quantitatively study tundish flow by obtaining the residence time distribution (RTD) curve through the stimulus-response method to calculate the ratio of the dead zone, piston zone, and fully mixed zone of tundish. The theoretical basis of the tundish hydraulic test is the similarity principle, and its basic condition is to ensure that the model is geometrically and dynamically similar to the prototype. The ratio of the geometric size of the model tundish used to that of the prototype tundish is 1:2. In terms of dynamic similarity, the content of the experiment is mainly related to the inertial force, gravity, and viscous force of the fluid in the tundish. According to the knowledge of fluid mechanics, when the molten steel flow in the tundish and the fluid flow in the water model are in the same self-modeling area, as long as the Fr numbers of the model and the prototype are equal, the dynamic conditions required for the experiment can be met. The experimental setup of the water model is shown in Figure 2.

During the water simulation experiment, the temperature of the water is about 20 °C, and its volume expansion coefficient is 2.07 × 10^−4^ K^−1^. For the simulated molten steel, its temperature is about 1500 °C, and its volume expansion coefficient is 38.3 × 10^−6^ K^−1^. In the tundish water model experiment, the water in the channel is heated to simulate the channel induction heating of the tundish, which makes up for the temperature difference between the inside and outside of the channel, which is about 10 °C. A single channel in the water model uses a 3 kW heating rod. Generally, the temperature difference at both ends of the heating channel in the tundish water model is about 2–3 °C so as to simulate the temperature difference at the channel position in the molten steel.

### 2.2. Mathematical Models

#### 2.2.1. Model Assumptions and Control Equations

Based on the COMSOL Multiphysics 6.0 software, this study analyzed in detail the impact of inclusion particles of various diameters being removed from the channel, the wall, and the top surface of the discharging chamber. To reflect the impact of induction heating’s presence or absence on the removal of impurities from the molten steel in the tundish, the following assumptions were made for the simulation.

Surface slag’s impact on flow is not taken into account;Molten steel is an incompressible fluid;The flow field is regarded as a stable flow field, and the inclusions are only affected by gravity, buoyancy, viscous resistance, and electromagnetic force;The collision growth of inclusions is ignored;If the inclusions contact the top slag, it means that they will be separated from the metal melt.

The Navie–Stokes equation, continuity equation, k-ε double equation, mass transfer equation, energy equation, and Maxwell’s equations [15,22] are used to establish a 3D numerical model of the tundish. When the inclusions move in the induction heating tundish, they will be affected by gravity, buoyancy, and Saffman lift. The equations used in the model are shown below.

In the calculation of tundish induction heating, gravity, buoyancy, drag force, and other forces are related to the flow rate of the molten steel so as to calculate the flow rate in different regions more accurately. The expression of the momentum equation is shown in Equation (1).
(1)$$\frac{\partial (u_{i}u_{j})}{\partial x_{j}}=-\frac{\partial P}{\partial x_{j}}+\frac{\partial }{\partial x_{j}}\left(\mu_{eff}\frac{\partial u_{i}}{\partial x_{j}}\right)+\frac{\partial }{\partial x_{i}}\left(\mu_{eff}\frac{\partial u_{j}}{\partial x_{i}}\right)+\rho g+F_{B}$$
where $x_{i}$ and $x_{j}$ are the space coordinate, m; P is the pressure, Pa; g and ρ are the acceleration of gravity and molten steel’s density, respectively, m·s^−1^ and kg·m^−3^; $\mu_{eff}$ is the effective viscosity coefficient, which is the sum of laminar flow viscosity and turbulent flow viscosity, kg·m^−1^·s^−1^; $F_{B}$ is thermal buoyancy, N.

The flow process in the continuous casting process is calculated using the low Reynolds number k-ε turbulence model, and the expression of the turbulent kinetic energy equation is shown in Equation (2).
(2)$$\frac{\partial (\rho u_{i}k)}{\partial x_{i}}=\frac{\partial }{\partial x_{i}}\left[\left(\mu_{eff}+\frac{\mu_{t}}{\sigma_{k}}\right)\frac{\partial k}{\partial x_{i}}\right]+G-\rho \varepsilon$$
where k means turbulent kinetic energy, m^2^·s^−2^; $u_{i}$ and $u_{j}$ are the velocity component in the $x_{i}$ and $x_{j}$ direction, m/s; ε represents kinetic energy dissipation rate, m^2^·s^−3^; $\mu_{t}$ means turbulent viscosity, kg·m^−1^·s^−1^; G means the turbulent kinetic energy source term.

The equations of turbulent kinetic energy dissipation rate are shown in Equations (3)–(5).
(3)$$\frac{\partial (\rho u_{i}\varepsilon )}{\partial x_{i}}=\frac{\partial }{\partial x_{i}}\left[\left(\mu_{eff}+\frac{\mu_{t}}{\sigma_{\varepsilon }}\right)\frac{\partial \varepsilon }{\partial x_{i}}\right]+c_{1}f_{1}G\rho \frac{\varepsilon }{k}-c_{2}f_{2}\rho \frac{\varepsilon^{2}}{k}$$
where $f_{1}$ = 1, $C_{1}$ = 1.45, $C_{2}$ = 2.0, and $f_{2}$ can be calculated by Equation (4).
(4)$$f_{2}=1-0.3exp(-{Re}^{2})$$
(5)$$G=\mu_{t}\frac{\partial u_{j}}{\partial x_{i}}(\frac{\partial u_{i}}{\partial x_{j}}+\frac{\partial u_{j}}{\partial x_{i}})$$

The values of the corresponding coefficients in the formula include $c_{\mu }$ = 0.09, $\sigma_{k}$ = 1.00, and $\sigma_{\varepsilon }$ = 1.00, which are recommended by Launder and Spalding [23].

The expression of the energy equation is Equation (6).
(6)$$\rho \left(\frac{\partial T}{\partial t}+C_{p}\frac{\partial T}{\partial x_{i}}\right)=\frac{\partial }{\partial x_{i}}\left(k_{eff}\frac{\partial T}{\partial x_{i}}\right)$$
where $k_{eff}$ is the effective thermal conductivity, W·m^−1^·K^−1^; T is temperature, K; $C_{p}$ is the specific heat capacity, J·kg^−1^·K^−1^.

The differential equation of force balance of inclusions in the tundish is shown in Equation (7). Due to the electromagnetic force, induction heating, and thermophoretic force in the tundish, the inclusion particles are affected by gravity, buoyancy, drag, and Staffman force in the tundish [24,25]. The motion equation of inclusion particles is as follows.
(7)$$\rho_{P}\frac{\pi }{6}d_{P}^{3}\frac{dv_{P}}{dt}=F_{g}+F_{b}+F_{d}+F_{s}+F_{p}+F_{t}$$
where $\rho_{P}$ is inclusions’ density, kg·m^−3^; $v_{P}$ represents the flow rate of inclusion particles, m/s; $d_{P}$ is the diameter of inclusion particles, μm; $F_{g}$ means the gravity of inclusion particles, N; $F_{b}$ represents the buoyancy force on the inclusion particles, N; $F_{d}$ is the drag force on the inclusion particles, N; $F_{s}$ means the Staffman force on the inclusion particles, N; $F_{p}$ is the electromagnetic pressure force on the inclusion particles, N; $F_{t}$ is the thermophoretic force on the inclusion particles, N.

#### 2.2.2. Boundary Conditions

Based on the actual situation, the outlet of the inclusion particles is set to the section of the outlet and the top surface in the pouring area, and the wall condition at the outlet is set to freeze. The conditions of the tundish wall and channel wall are set to rebound, and the condition of the primary particle condition is probability 0.5; otherwise, the inclusion particles freeze on the wall. The induction coil used in this study is a pair of single-phase alternating-current windings with a frequency of 50 Hz and a power of 800 kW. The theoretical residence time of the tundish is 1080 s, and double the theoretical residence time, 2160 s, is input into the mathematical model of the motion trajectory of the inclusion particles, providing sufficient time for the removal of the inclusion particles and solving the state of inclusion particles at different times through a transient solver.

Table 1 indicates the properties of molten steel together with parameters in mathematical modeling.

### 2.3. Mesh Independent Study

Since the research focus of this paper is the molten steel in the tundish, only grid-independent research is done on the tundish. The mesh division of the induction heating tundish is exhibited in Figure 3.

The grid division of the tundish is all free tetrahedral grids, and the positions of the channel connection, inlet, and outlet have been refined, and five layers of boundary layer grids have been set to improve the accuracy of settlement results. The results of grid-independent research on induction heating tundish are shown in Table 2, with a total number of 1.3 million grids.

The tundish’s average flow rate of molten steel is 0.0209 m/s, and the average flow velocity is kept with the number of grids increasing. Therefore, the optimal solution mesh number of the tundish three-dimensional (3D) model should be 1.3 million. Based on this, the number of tundish grids established in this study is 1,328,585.

### 2.4. Model Validation

Since the research mechanism of Vives et al. [26] is similar to the induction heating mechanism involved in this paper, the experimental results of Vives et al. are selected for verification. The comparison results are shown in Figure 4.

The magnetic field strength of the simulation results of this study and the results of Vives et al. [26] are both 0.04 T on the Z = −20 cm section of the left and middle channels. The intensity and direction distribution of the magnetic field are similar, and they all have the characteristics of a symmetrical magnetic field in the middle channel section. The magnetic field of the center channel is distributed symmetrically, and the magnetic field of the left channel is distributed eccentrically due to the proximity effect. It can be seen that the predicted results are in good agreement with the experimental data, which show the reliability of the simulation results of this model.

## 3. Results and Discussion

### 3.1. Physical Simulation Results

Due to the symmetry of the T-shaped tundish, only the RTD curves and flow field change trends of outlet 1, outlet 2, and outlet 3 are analyzed. The RTD curves for the three outlets are shown in Figure 5.

It can be seen from Figure 5 that the RTD curve obtained by numerical simulation is in good agreement with the RTD curve obtained by physical simulation, which shows the reliability of the model in this paper. Outlet 1 and outlet 3 have faster response times than outlet 2. The consistency of stream 1 and stream 3 is high, and the concentration of none in outlet 2 is relatively low. The reason is that the molten steel passes through outlet 1 and outlet 3 after flowing out from the branch, then hits the inner wall of the tundish, the relatively flowing molten steel forms a vortex, and then part of the molten steel flows to the outlet 2. Due to various changing factors in the experimental process during actual detection, the measured curve is tortuous, but its trend is still consistent with the simulation results.

### 3.2. Mathematical Simulation Results

#### 3.2.1. Simulation of the Temperature and Flow Fields

Compared with the traditional tundish, the induction heating tundish adopts an external heating source to keep molten steel’s temperature constant during pouring. This technology has a great influence on the flow field, temperature field, and behavior characteristics of inclusions in the tundish.

The flow field and temperature field distribution in the molten steel are described through the horizontal section of the tundish at the height of the channel center. Figure 6 shows the schematic diagram of the horizontal section of the tundish at the height of the channel center.

The flow and temperature field of molten steel on the center height section of the channel are shown in Figure 7 and Figure 8.

As depicted in Figure 7, the molten steel travels from the two sub-channels of the channel to the pouring region after passing through the channel. According to the principle of the continuity equation, the flow velocity gradually decreases after flowing out of the sub-channels. Additionally, the molten steel forms a large vortex in the variable flow area of the pouring area (outlet 1 and outlet 6) and forms a small vortex at the middle nozzle position (outlet 2 to outlet 5). This phenomenon contributes to the extension of the molten steel’s residence duration in the tundish, the decrease of the dead zone’s volume percentage, and the complete mixing of the molten steel’s constituent parts. The molten steel in the channel in Figure 8 is heated due to induction heating. After the molten steel flows through the left and right channels, the temperature rises from 1853.0 K and 1853.1 K to 1863.0 K and 1863.3 K, and the temperature increases by 10.0 K and 10.2 K, respectively. The average temperature of every outlet is shown in Table 3.

Each outlet has a maximum temperature variation of 5.5 K. Because outlet 2 and outlet 5 are located between two sub-channels, the molten steel flowing from the sub-channels needs to go through a long path to reach these two water outlets. Therefore, the average temperature of the sections of outlet 2 and outlet 5 is relatively low. The analysis shown above reveals that the temperature distribution in the channel could be raised greatly using induction heating technology. In addition, the application of induction heating helps to improve the cooling of molten steel at the initial stage of pouring so that the molten steel can be poured with low superheat.

#### 3.2.2. Simulation of Inclusion Removal from Tundish with and without Induction Heating 

On the effectiveness of removing inclusions from the tundish with and without using induction heating technology, a simulated investigation was carried out, and Table 4 displays the outcomes. Inclusions particles have a diameter of 50 μm.

In the receiving chamber, where inclusion removal mostly occurs, the removal rate is 97.5% without induction heating and 96.4% with it. Compared with the condition without induction heating, the adsorption rate of inclusion particles in the channel increases under the condition of induction heating, which indicates that the induction heating technology is helpful to the removal of inclusion in the channel. The simulation calculation results of the movement trajectories of inclusion particles in the channel and discharging chamber illustrate that the inclusion particles are completely removed within 346 s without induction heating; the inclusion particles were completely removed within 302 s when induction heating was used; the time was shortened by 44 s and decreased by 12.72%. This phenomenon further shows that induction heating technology is helpful in promoting the movement and removal efficiency of inclusions.

#### 3.2.3. Simulation of Inclusion Removal with Various Sizes Using Induction Heating 

Only the motion distribution of inclusion particles in the channel region and pouring area is simulated so as to analyze the removal of inclusion particles there thoroughly. Based on the original model, the entrance of the channel is set as the entrance of inclusion particles. The inclusion particles with diameters of 10 μm, 30 μm, and 50 μm in the channel and discharging chamber were simulated, and the trajectory of inclusion particles under different times was intercepted, as shown in Figure 9, Figure 10 and Figure 11.

Among them, the blue particles represent the inclusion particles, and the curve represents the trajectory of the inclusion particles. Figure 9, Figure 10 and Figure 11 show that the flow characteristics of inclusion particles with different diameters are similar, they are all adsorbed in a large amount in the channel, and only a small amount of inclusion particles flow to the discharging chamber.

The inclusion particles are released from 0 s, and the flow time of the inclusion particles in the channel is about 7 s. At 9 s, the inclusion particles first flow out from the sub-channel opening near the center, and some of the inclusion particles are adsorbed on the wall of the pouring area, as shown in the blue circle. The remaining particles continue to move, such as the particles in the circle at 112 s. Among them, most inclusion particles with a diameter of 10 μm were adsorbed at the channel, and the removal rate was 70.9%. A small part flowed to the discharging chamber and adhered to the wall of the discharging chamber. With a removal rate of 20.7%, the removal rate at the top surface of the discharging chamber was 4.2%, and no inclusion particles were detected at the outlet. Most inclusion particles with a diameter of 30 μm adhered to the channel, and the removal rate was 60.9%. Followed by the pouring area wall, the removal rate was 26.1%, 7.6% of the inclusion particles floated to the pouring area liquid surface, and no inclusion particles were detected at the nozzle. The inclusion particles with a diameter of 50 μm are mostly removed in the channel area, with a removal rate of 56.1%, followed by the wall of the pouring area, with a removal rate of 31.8%. A total of 7.9% of the inclusion particles floated to the top surface of the discharging chamber, and no inclusion particles were detected at the nozzle. According to the movement trajectories of inclusion particles, it can be seen that inclusion particles of 10 μm, 30 μm, and 50 μm were removed in 226 s, 344 s, and 368 s, respectively.

The removal effect and removal time of three kinds of inclusion particles with different diameters in each region are displayed in Figure 12. From Figure 12a, when the diameter of inclusion particles increases from 10 μm to 50 μm, the clearance rate of particles in the channel steadily declines from 70.9% to 56.1%. The clearance rate of particles on the wall of the discharging chamber increased from 20.7% to 31.8%, the clearance rate of particles at the top surface of the discharging chamber increased by 3.7%, and the removal rates of particles with three diameters at the nozzle were all 0%. The above analysis shows that the inclusion particles with smaller diameters are more likely to be adsorbed on the wall of the channel, and the inclusion particles with larger diameters are more likely to adhere to the wall of the pouring area or float up to the top surface of the discharging chamber.

Figure 12b shows that when the particle diameter continues to increase, the removal time of inclusion particles shows an overall upward trend. When the particle diameter is 10 μm, the particles are removed within 226 s, and when the particle diameter of inclusions is 50 μm, the removal time is extended to 368 s. The clearance time of particles in the pouring area is connected to their floating speed and particle concentration in molten steel. The expression of the floating velocity of inclusion particles is shown in Equation (8).
(8)$$U_{p}=\frac{(\rho_{l}-\rho_{p})\cdot g\cdot D^{2}}{18\mu }$$
where $U_{p}$ is the floating speed of inclusion particles, m/s; $\rho_{l}$ and $\rho_{p}$ represent the density of steel and inclusions, respectively, kg/m^3^; μ is the viscosity of molten steel, Pa·s; D means the diameter of inclusions, μm.

According to Equation (8), for the same kind of inclusions, the floating speed of the inclusion particles increases as the inclusions’ diameter steadily rises, which is advantageous for the inclusion particles’ removal. The simulation results under induction heating conditions show that the clearance rate of particles with small sizes in the channel area is obviously higher than that of particles with large sizes. When the total number of particles released at the entrance remains unchanged, the concentration gradient of particles with large diameters in the pouring area is higher than that with small diameters among the particles flowing into the pouring area. As a result, inclusion particles with larger diameters take longer to remove from the discharging chamber than particles with smaller diameters.

The distribution and removal of inclusion particles in tundish have always been a hot issue in tundish metallurgy. However, due to the sealing and danger of field operation, it is difficult to detect the distribution of inclusion particles in tundish with field data, which is why this paper does not verify the distribution of inclusion particles in various areas of tundish with factory data. Subsequently, the influence of different parameters on the removal of inclusion particles in tundish will be further studied, and comparative analysis will be carried out through the field data of inclusion particles as far as possible.

## 4. Conclusions

The mathematical simulation results of the removal of inclusions with and without induction heating show that the induction heating technology contributes to promoting the movement and removal of inclusions in the channel. Meanwhile, induction heating technology shortens inclusion particle removal time;Smaller inclusion particles are easier to be adsorbed in the channel, and larger inclusion particles are easier to be removed in the discharging chamber. The clearance rate of inclusion in the channel gradually decreases from 70.9% to 56.1%, with the diameter of inclusion particles increasing from 10 μm to 50 μm;The inclusion particles with large particle sizes are easier to be removed on the wall and liquid surface of the discharging chamber. Moreover, compared with small-size inclusions, the removal time of large-size inclusions is longer.

## Figures and Tables

**Figure 1 materials-16-05254-f001:**
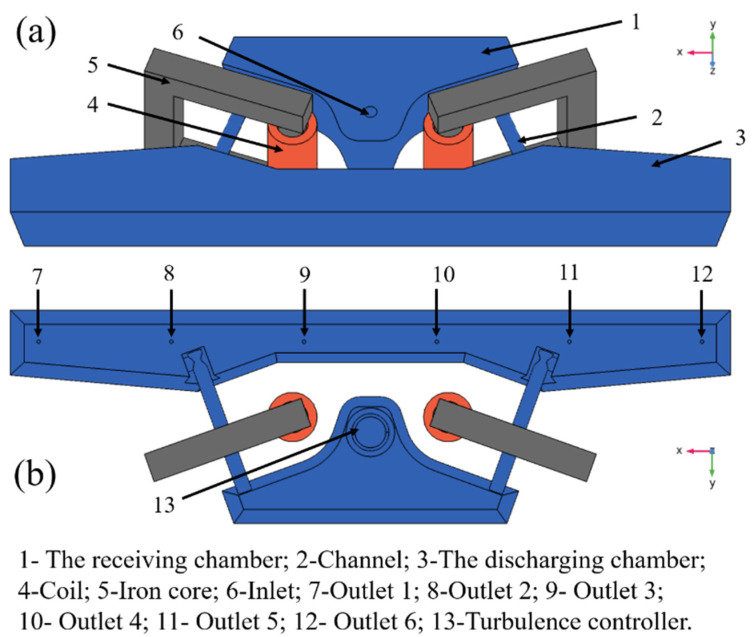
Dual-channel electromagnetic induction heating tundish model: (**a**) upper view; (**b**) bottom view.

**Figure 2 materials-16-05254-f002:**
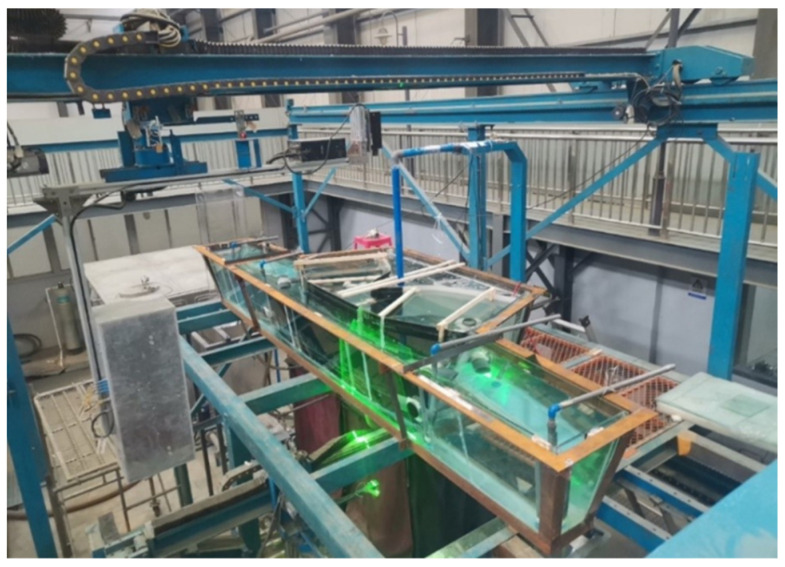
Six-flow tundish water model experimental device.

**Figure 3 materials-16-05254-f003:**
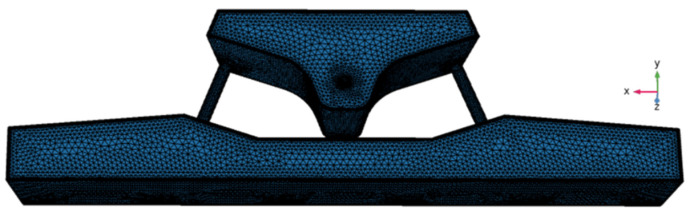
Mesh division diagram of induction heating tundish.

**Figure 4 materials-16-05254-f004:**
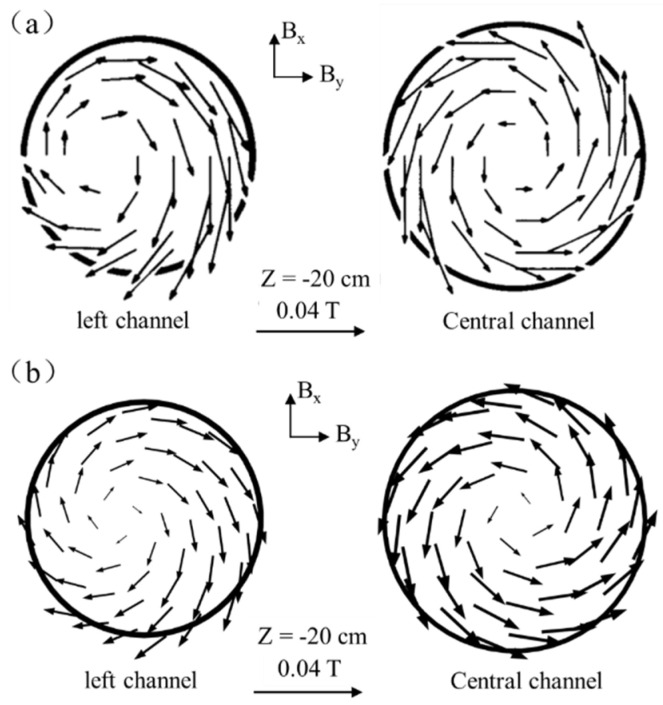
Comparison of magnetic field intensity distribution: (**a**) experimental results; (**b**) simulation results of this model.

**Figure 5 materials-16-05254-f005:**
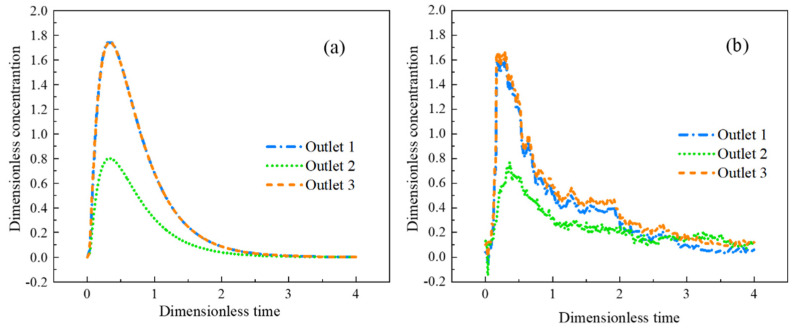
Comparison of simulated and physical-simulated RTD curves: (**a**) numerical simulation results; (**b**) physical simulation results.

**Figure 6 materials-16-05254-f006:**
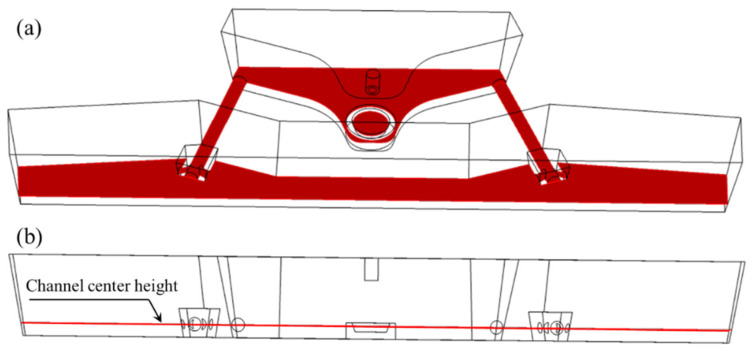
Schematic diagram of channel center height section: (**a**) upper view; (**b**) front view.

**Figure 7 materials-16-05254-f007:**
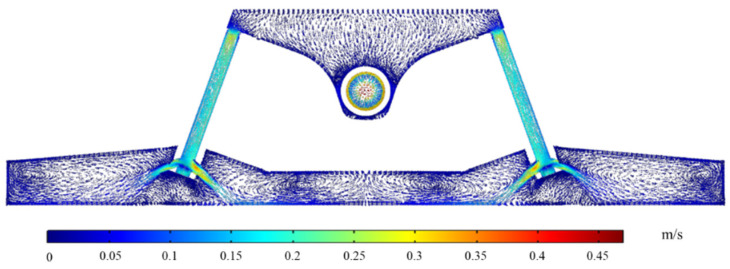
Flow field distribution on the center height section of tundish channel.

**Figure 8 materials-16-05254-f008:**
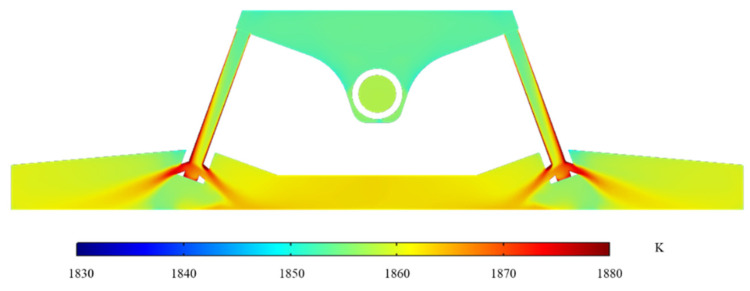
Temperature field distribution on the center height section of tundish channel.

**Figure 9 materials-16-05254-f009:**
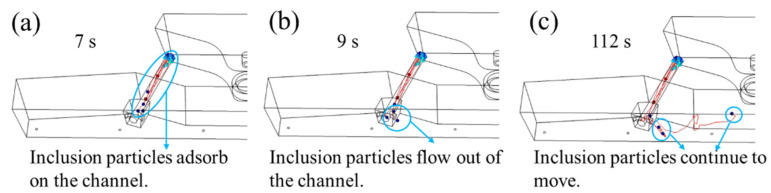
Trajectories and removal effects of 10 μm inclusion particles at (**a**) 7 s; (**b**) 9 s; (**c**) 112 s.

**Figure 10 materials-16-05254-f010:**
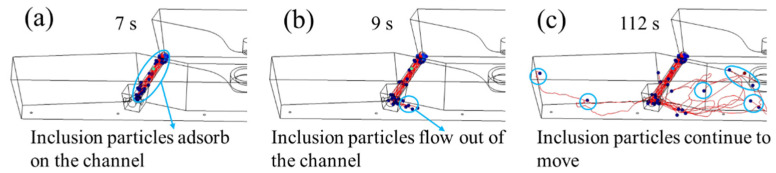
Trajectories and removal effects of 30 μm inclusion particles at (**a**) 7 s; (**b**) 9 s; (**c**) 112 s.

**Figure 11 materials-16-05254-f011:**
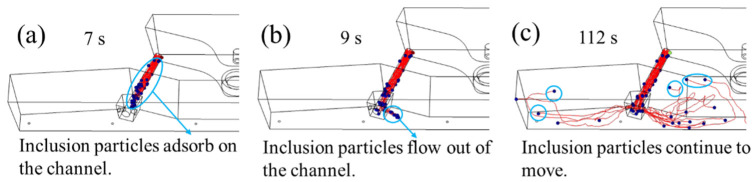
Trajectories and removal effects of 50 μm inclusion particles at (**a**) 7 s; (**b**) 9 s; (**c**) 112 s.

**Figure 12 materials-16-05254-f012:**
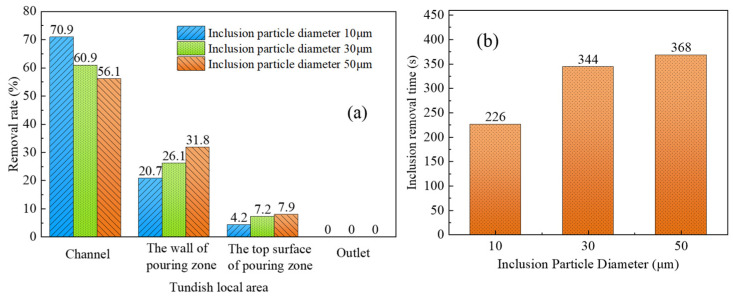
(**a**) Removal rate and (**b**) removal time of inclusion particles with different diameters.

**Table 1 materials-16-05254-t001:** Steel physical parameters and simulation parameters.

Parameter	Value	Parameter	Value
Inlet velocity, m·s^−1^	0.9	Thermal conductivity, W·m^−1^·K^−1^	23.5
Inlet temperature, K	1833	Heat capacity at constant pressure, J·kg^−1^·K^−1^	4500
The diameter of the particle, μm	10, 30, 50	Dynamic viscosity, Pa·s	0.0065
Particle density, kg·m^−3^	3900	Surface heat loss, W·m^−2^	15,000
Particle count released at a time	1000	Bottom heat loss, W·m^−2^	1800
Particle precision order	5	Side heat loss, W·m^−2^	4600
Molten steel density, kg·m^−3^	7580	Channel heat loss, W·m^−2^	2000

**Table 2 materials-16-05254-t002:** Grid-independent calculation results of induction heating tundish.

Number of Grids	400,000	600,000	1,000,000	1,300,000	1,800,000	2,000,000
Average velocity, m·s^−1^	0.0194	0.0198	0.0203	0.0209	0.0209	0.0209

**Table 3 materials-16-05254-t003:** The average temperature of each outlet.

Outlet’s Number	1	2	3	4	5	6
Average temperature, K	1860.3	1857.1	1862.3	1862.5	1857.0	1860.5

**Table 4 materials-16-05254-t004:** Removal rate of inclusions with and without induction heating.

Local Area in the Tundish	Removal Rate without Induction Heating, %	Removal Rate with Induction Heating, %
The receiving chamber	97.500	96.400
Channel	2.100	3.000
The wall of discharging chamber	0.030	0.040
Liquid level in discharging chamber	0.000	0.000
Outlet	0.000	0.010

## Data Availability

Data are contained within the article.

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
