# Peer review of "The Removal of Inclusions with Different Diameters in Tundish by Channel Induction Heating: A Numerical Simulation Study"

_materials, 2023, doi:10.3390/ma16155254_

Round 1
Reviewer 1 Report
Abstract, line 16: Abbreviation RTD should be fully written. If an external reader only reads the abstract, this abbreviation is not clear at first sight. Abbreviation not clear for external reader if she/he only reads the abstract without going deeper into the paper. Write full meaning (Residence Time Distribution) into the abstract.
Page 3, line 99. Insert additional sentence "Following assumptions were made for the simulation" and put this before the list of boundary conditions. Insert the text part put in brackets into the manuscript before the bullet points with the listed boundary conditions. This information helps external readers to better understand the simulation set-up.
Page 5: A few comments about the convergence criteria should be inserted. When does the simulation stop, i.e., when do you know if the calculation results are steady? Is there any parameter, which should reach a certain value to show you when the simulation is finished? E.g., a certain temperature level? Or do you stop the calculation when the values do not change anymore over a certain time?
Conclusions: is further research planned on that topic? If yes, what are the future challenges. A further comparison with industrial plant data would be valuable in this regard, since this could validate the quality of the simulations. Could you comment on this? Do you plan further research on that topic? If yes, what are the challenges and will there be the chance to validate with data from industrial plants?
Reviewer 2 Report
Review of the work “ The Removal of Inclusions with Different Diameters in Tundish by Channel Induction Heating: A Numerical Simulation Study”
The article deals with the physical and mathematical modeling of inclusion removal in Tundish with channel induction heating
In the introductory part, the bibliographical review is incomplete since it omits recent articles directly related to the topic under study, for instance, the following reference:
Wang, P., Xiao, H., Chen, X. Q., Li, X. S., He, H., Tang, H. Y., & Zhang, J. Q. (2021). Influence of dual-channel induction heating coil parameters on the magnetic field and macroscopic transport behavior in T-type tundish. Metallurgical and Materials Transactions B, 52, 3447-3467.
The authors forgot to include in the modeling section the detailed description and justification of the physical model and the experimentation carried out with it to arrive at the results shown in Figure 4, which are used to validate the numerical model.
Instead of the above in section 2.1 physical model, the authors present the schematic diagram of the system's structure under study and its meshing for its numerical modeling using fluent.
In addition, the explanation of the system under study represented schematically in Figure 1 is deficient, see for example, in lines 63 and 64 of page 2, what is said about this figure: “ The long nozzle is where the molten steel enters, passes through the pouring zone, the channel and the pouring zone, and finally flows out from the outlet.”
The validation of the distribution of the intensity of the magnetic fields shown in Figure 3 is not clearly explained in the text since the authors do not clarify to the reader that it is the simulation of Vives's experimental system applying the numerical model elaborated by the authors.
In the simulations carried out with the numerical model, the parameters of the induction coil are not specified.
The work presents many deficiencies, such as those listed above, which prevent it from being considered for publication in Materials.
I suggest authors take into account the following advice to rewrite their article before submitting it to this or another journal for publication:
Rewrite the introduction considering all the works published to date about the simulation of removal of inclusions in continuous casting Tundish with channel-type induction heating in such a way that it shows state of the art in this subject and the originality of the work done by the authors.
At the beginning of the model construction section, include an explanation of how the system under study works, from the moment the metallic stream enters from the ladle until it leaves the tundish and is delivered to the continuous casting molds through the six outlets. Use this explanation to describe Figure 1, including the presence and location of the induction coils and commenting on the effects of the induced electromagnetic field on the liquid flow through the inlet channels.
In the model construction section, explain the physical model and the experimentation carried out based on compliance with similarity criteria between the model and the prototype.
Explain in greater detail and clarity the validation of the numerical model with the results of the experimentation with the physical model, as well as the validation of the predictions of the intensity of the magnetic field.
Once the validation of the model has been shown to the reader, comment promptly and clearly on the modeling strategy that was followed to simulate the removal of inclusions of different sizes and what is intended to be analyzed.
The power used in the inductor significantly affects the removal of inclusions; for this reason, the authors must choose for their simulations induction coil parameters to show more clearly the effects of the induced electromagnetic field on the removal of inclusions.
Finally, and to avoid the current confusion that is created by naming the pouring zone, both the part of the tundish receiving the metal stream from the ladle and the zone of the tundish delivering the liquid metal to the continuous casting molds through the six outlets I suggest the following terminology: That the part of the tundish that receives the metal jet from the ladle be called the receiving chamber and that the zone of the tundish delivering the liquid metal to the continuous casting molds through the six outlets be called the discharging chamber, and that these denominations be used throughout the entire article.
Reviewer 3 Report
1. Kindly include more recent literature that relates to the work to justify the relevance of the proposed work. Most of the references are more than 5 years with only 4 references are within 5 years.
2. Include preamble for
a. Section-2 to introduce its subsection of 2.1 and 2.2.
b. Subsection-2.1 to introduce its 2nd level subsection of 2.1.1 and 2.1.2
c. Subsection-2.2 to introduce its 2nd level subsection of 2.2.1 and 2.2.2
d. Section-3 to introduce its subsection of 3.1 and 3.2.
e. Subsection-3.2 to introduce its 2nd level subsection of 3.2.1, 3.2.2 and 3.2.3
3. The mesh convergence study from line 69 until 81 can be moved to section 2.3 such,
a. Section “2.3 Mesh Independent study”
b. Section “2.4 Model Validation”
Hence, section “2.1.1 Geometry”
4. How does the Dual-channel electromagnetic induction heating tundish model is idealised as given in Figure-1? Kindly include this information to the manuscript.
5. Kindly include the figure for the physical setup of the experiment as described in section 2.1.2
6. It is highly suggested that Figure-4 to use different grayscale or line type for the plotting rather than use a different colour.
7. Further elaboration on the Figure-4 is required especially on the profile/trend for each of the outlets.
8. Kindly label the axes with a proper name. Such C/Co represents to what kind ration and theta represent to what kind of direction.
9. Referring to Table-4, the “Number of outlets” statement can be misleading, hence use “Outlet’s number” instead. Also kindly mentioned after line 208, the outlet’s number can be referred to the previous figure of Figure-1.
10. It is suggested the conclusion to be more conclusively rather than describing the parametric results such on what is the most preferable inclusion particle diameter and how it is being selected?
English is acceptable and understandable
Round 2
Reviewer 2 Report
I consider that the corrected version sent by the authors has adequately addressed the reviewers' comments and that, in its current state, it presents a very good work. This work interests all readers interested in metal alloy ingot production by continuous casting and specifically in improving the metallurgical quality of steels through better designs and technologies used in the tundishes to control inclusion content in the metallic stream that feed the continuous casting machines. Therefore, I consider that this work should be published in Materials.